# Anxiety Sensitivity Social Concerns Predicts Electrodermal Activity during the Niacin Biological Challenge Paradigm

Kevin G. Saulnier [1,2,*] , Marija Volarov [3] and Nicholas P. Allan [4]

1 Department of Psychology, Ohio University, Athens, OH 45701, USA
2 Department of Psychiatry and Behavioral Health, Penn State College of Medicine, Hershey, PA 17033, USA
3 Department of Psychology, Faculty of Sport and Psychology, Educons University, 21208 Novi Sad, Serbia
4 Wexner Medical Center, Ohio State University, Columbus, OH 43210, USA
* Correspondence: ks981615@ohio.edu

**Abstract:** Anxiety sensitivity social concerns (ASSC), or the fear of observable anxiety symptoms, is a risk factor for social anxiety. ASSC predicts anxiety following the niacin biological challenge, a paradigm in which niacin is used to manipulate facial flushing during a speech task. However, it remains unclear if ASSC predicts physiological arousal during this task. The current study was designed to examine the effects of ASSC on self-reported distress and electrodermal activity (EDA) during the niacin biological challenge in a sample of undergraduates ($N = 36$; $M$ age = 18.9, $SD$ = 0.84; 69.4% female). Participants were randomly assigned to one of four conditions in a 2 (100 mg niacin vs. 100 mg sugar) $\times$ 2 (instructional set) design. Participants completed a speech task in a virtual reality environment. Participants rated their distress halfway through the speech and EDA was averaged over four intervals. There was a main effect for ASSC on subjective distress. There was a significant ASSC by condition interaction predicting EDA, in that ASSC was related to EDA only in the niacin condition. ASSC also was more strongly related to EDA anticipating the speech. These findings highlight the role of ASSC in predicting anxiety and physiological arousal.

**Keywords:** social anxiety; anxiety sensitivity; electrodermal activity; virtual reality

## 1. Introduction

Social anxiety disorder (SAD) is characterized by maladaptive distress and avoidance in social situations [1] and represents a substantial public health burden [2]. Thus, it is important to identify risk factors for social anxiety. One such risk factor is anxiety sensitivity social concerns (ASSC), which is the fear of publicly observable symptoms of anxiety (e.g., blushing, trembling, sweating) [3]. People who score high on ASSC have irrational beliefs that these symptoms will lead to social rejection or public humiliation [3]. Meta-analytic studies support the link between ASSC and social anxiety symptoms and diagnoses of SAD [4]. Further, ASSC predicts social anxiety above the influence of other anxiety sensitivity dimensions [5] and ASSC predicts panic symptoms following social stressors in daily life [6]. However, few studies have explored the role of ASSC in the development of acute social anxiety using experimental designs. Experimental designs conducted in the laboratory can provide evidence to make the strongest possible causal inferences regarding the processes through which ASSC amplifies anxiety during social situations.

There are several experimental approaches for anxiety sensitivity generally [7–9] and several interoceptive exposure techniques for individuals with social anxiety [10]; however, fewer experimental methods have been developed specifically for ASSC. One experimental approach to assess how ASSC impacts social anxiety during social situations is through the use of biological challenge paradigms, or paradigms that involve the manipulation of feared sensations through the use of a biological mechanism. To date, niacin (Vitamin B3), which induces facial flushing through the activation of vasodilatory prostaglandin receptors [11], has been used as a biological agent in a biological challenge paradigm to

measure ASSC [12]. In this approach, participants are randomly assigned to consume niacin or a sugar pill placebo prior to completing a speech task in which they are told they will be judged based on their publicly observable symptoms of anxiety. In the only previous study to employ this paradigm, ASSC predicted self-reported panic symptoms that were assessed following the speech, and this effect was specific to the niacin condition [12]. However, ASSC was not related to subjective distress (measured by having participants report their acute distress on a 1–100 scale) captured following the speech, potentially due to participants having sufficient time for subjective distress to remit following the challenge combined with only a moderately distressing challenge (e.g., participants were not given feedback based on their observable anxiety symptoms).

Allan et al. [12] provided initial support for the niacin biological challenge as a paradigm to assess ASSC. However, they only examined self-reported distress. Additionally, there have been no published manuscripts designed to replicate the results from Allan et al. [12]. These findings need to be replicated conceptually, ideally across multiple units of analysis to address the potential of self-report bias [13,14]. Further, in the Allan et al. [12] study, self-reported distress was collected *after* the speech. As Hellhammer and Schubert [15] found, the covariation between self-report and alternate units of analysis (i.e., cortisol, heart rate) was optimal when both were measured during a speech. In contrast, self-report and physiological measures did not covary before or after the speech. Therefore, to capture acute anxiety in the lab, it may be important to assess subjective distress *during* the speech as opposed to *following* the speech when administering the niacin biological challenge. This approach is consistent with how distress is measured during exposure therapy, by getting ratings of subjective distress during exposures [16].

When selecting physiological indicators, it is ideal to select indicators that are empirically and theoretically relevant to the experience of anxiety. The NIMH Research Domain Criterion [17] includes electrodermal activity (EDA) as a physiological indicator of acute threat response. EDA is a measure of sweat gland secretion widely used in laboratory-based social stress tasks as a marker of physiological arousal [18]. Clarifying whether ASSC impacts EDA when giving a speech provides support for ASSC as impacting not only perceived distress but the modulating influence of ASSC on physiological arousal during a social stressor.

The current study was designed to evaluate the effects of ASSC on anxiety reactions during an experimental manipulation of publicly observable anxiety symptoms in a social setting. In the current study, participants were randomized to consume either niacin or sugar (placebo) before completing a speech task in a virtual reality environment (a virtual conference room with an audience). Several changes were made from the Allan et al. [12] study to replicate and extend their findings. First, a virtual reality audience was used in the current study, as opposed to trained confederates as used in the Allan et al. [12] study, to ensure consistent observer reactions across participants. Virtual reality has been shown to reliably elicit physiological stress responses, including EDA, in a recent meta-analysis of laboratory studies [19]. Anxiety reactions during the niacin biological challenge paradigm were assessed in several novel ways, namely by having participants rate their subjective distress *during* the speech task and by assessing tonic EDA throughout the speech task. Like the Allan et al. [12] study, panic ratings were also obtained after the speech task. It was expected that ASSC would be positively related to all forms of anxiety reactions, particularly in the niacin condition, consistent with previous research [12]. Finally, the influence of ASSC on EDA trajectory throughout the speech was examined. Given ASSC is conceptualized as an amplifier of anxiety symptoms, it was expected that the impact of ASSC on EDA would be stronger later in the speech. By validating the use of niacin as a biological challenge relevant to ASSC, confidence is this protocol as a paradigm to investigate processes that contribute to social anxiety will be increased, ultimately leading to a better understanding of the etiology of social anxiety.

## 2. Materials and Methods

### 2.1. Participants

Participants were 36 undergraduates recruited through the subject pool at Ohio University between December 2016 and December 2019. All undergraduate students enrolled in the introductory psychology courses were automatically entered into the subject pool and completed a battery of screening questionnaires at the beginning of each semester. Participants were then able to register to participate in research studies for course credit. Access to this study was limited to participants who reported at-least mean levels of overall AS (i.e., a mean score of 13 on the Anxiety Sensitivity Index-3) [3]. Exclusionary criteria for this study included a niacin allergy, diabetes, a bleeding disorder, kidney disease, gout, liver dysfunction, liver disease, peptic ulcer disease, arterial bleeding, hypotension, and rosacea. Additionally, participants were excluded if they had recently had surgery or planned to have surgery in the next two weeks, were pregnant or breastfeeding, or were taking medications that affected their blood sugar or increased their risk of bleeding. Exclusionary criteria were assessed by self-report of the participant. No participants were excluded based on these criteria.

### 2.2. Measures

**Anxiety Sensitivity Index-3 (ASI-3).** The ASI-3 is an 18-item self-report measure designed to assess anxiety sensitivity [3]. The ASI-3 contains three subscales; only the ASSC subscale was used in the current study to capture trait ASSC. The three-factor structure of the ASI-3 has been confirmed across several studies [3,20–23]. The ASI-3 was developed using an English-speaking university sample and has demonstrated strong reliability and validity metrics among university students [3]. Further, the ASSC subscale has been found to have good convergent and divergent validity [3,5,20,24]. Finally, the ASSC subscale of the ASI-3 is elevated among individuals diagnosed with SAD, and individuals with SAD report higher levels of ASSC relative to the other dimensions of anxiety sensitivity [3,23,24], supporting the construct validity of this measure. Internal consistency for this subscale in this sample was excellent ($\alpha = 0.90$; 6 items).

**Acute Panic Inventory (API).** The API is a 24-item self-report questionnaire designed to measure panic symptoms [25]. The API was used in the current study to measure in-the-moment panic symptoms. The API has demonstrated strong reliability across other fear provocations studies that used English-speaking university samples [9,12,26]. Further, the API is sensitive to acute changes in response to stressors [27,28]. Finally, individuals with panic disorder and other anxiety disorders consistently report higher scores on the API following acute stress relative to individuals without anxiety disorders [29,30], supporting the construct validity of this measure. Internal consistency for the API in the current study was excellent ($\alpha = 0.92$).

### 2.3. Procedures

**Laboratory Tasks.** Upon arrival to the lab, participants provided informed consent. During the informed consent process, all participants were informed that they may consume a vitamin $B_3$ supplement and that this supplement could cause slight facial flushing. Participants were then randomized to one of four conditions in a 2 (drug) $\times$ 2 (instructional set) factorial design. Participants were randomly assigned to one of the two drug conditions (niacin or placebo) and to one of the two instructional set conditions (control or increased salience condition). Regarding drug condition, participants were randomly assigned to receive either 100 mg of niacin or 100 mg of sugar (placebo) in a double-blind fashion using identical gelatin capsules. Participants were then affixed to physiological equipment and completed a battery of self-report measures and cognitive tasks (lasting approximately 45 min) to allow for niacin absorption. Participants then were given instructions for the speech task. Participants were also attached to a vibrating finger cuff and were given differing instructions depending on instructional set condition. Participants assigned to the control condition were told that the finger cuff would vibrate randomly throughout the

speech task, whereas participants assigned to the increased salience condition were told that the finger cuff would vibrate when their blushing reached a certain level. In reality, the finger cuff vibrated in a fixed pattern for all participants.

Before beginning the speech, participants completed a one-minute baseline recording for physiological data. Participants were then instructed to complete a five-minute speech on an anxiety provoking topic (describing their greatest weaknesses). Participants gave these speeches to a virtual audience using an Oculus Rift headset. The Fiboni Presentation Simulator [31] was used to simulate a conference room environment and to standardize audience appearance across participants. Participants were stopped halfway through the speech for ten seconds to obtain ratings of subjective distress (on a 1–10 scale). If participants were silent for more than 10 s, they were prompted to continue their speech until their time was up. After the speech participants completed a brief battery of self-report questionnaires and were debriefed before leaving the laboratory.

**Physiological Recording**. Tonic EDA was recorded using the BIOPAC MP160 EDA data acquisition module and two adhesive Ag/AgCl electrodes (Biopac EL507) with iso-tonic gel affixed to the pointer and middle fingers. An online low pass filter was applied at 1.0 Hz, and 5 μS/V of online gain was applied. Tonic EDA was averaged over the following time intervals: the minute before the speech (pre-speech EDA), the first 2.5 min of the speech (early-speech EDA), the 10-s pause in the middle of the speech (pause EDA), and the last 2.5 min of the speech (late-speech EDA).

*2.4. Data Analytic Plan*

The main statistical analyses were conducted in SPSS version 28 [32]. First, descriptive analyses were conducted to examine violations of normality. Missing data were handled using listwise deletion (i.e., participants with missing values in at least one of the specified variables were excluded for that analysis). Bivariate correlations were also estimated for all study variables. Separate linear regression models were estimated to examine predictors of subjective distress reported during the speech task and panic symptoms reported following the speech. In these linear regression models, the effects of ASSC, drug condition (dummy coded such that 1 = niacin condition), instructional set condition (dummy coded such that 1 = increased salience condition), the ASSC by drug interaction, and the ASSC by instructional set interaction were entered simultaneously. This same approach was taken to examine predictors of subjective distress during the speech and panic symptoms following the speech. Finally, a repeated-measures analysis of variance (ANOVA) was conducted to examine the effects of ASSC, drug condition, instruction condition, the ASSC by drug interaction, and the ASSC by instruction interaction on EDA over time. Average tonic EDA was treated as a within-person variable and was assessed across pre-speech, early-speech, pause, and late-speech time windows. Across all analyses, nonsignificant interactions were removed from the final models for ease of interpretation.

Power analyses were conducted to determine the sample size needed to detect large effects for the main parameters of interest using G*Power. For the linear regression analyses, the power required to detect an $R^2$ increase in a fixed model was determined for the two main parameters of interest (ASSC by drug interaction, ASSC by instructional set interaction) in a model with five total predictors. Specified parameters included a large effect size ($F = 0.35$) with two-tailed $\alpha = 0.95$ and power $(1 - \beta) = 0.80$. This power analysis indicated that a total sample of 32 would be sufficiently powered to detect a large effect size. For the repeated-measures ANOVA, power analyses were conducted to determine the power required to detect between-person effects, within-person effects, and the interaction of within- and between-person effects. A large effect size was used for all analyses ($F = 0.35$) with two-tailed $\alpha = 0.95$, and power $(1 - \beta) = 0.80$. The power was determined to detect differences between two groups with four measurements within each group. A correlation of 0.70 for repeated measures was used. Given these parameters, a sample of 52 would be sufficiently powered to detect between-group effects, a sample of 10 would be sufficiently powered to detect within-person effects, and a sample of 10 would

be sufficiently powered to detect within- and between-person interactions. Therefore, the repeated-measures ANOVAs presented in this manuscript were likely underpowered to detect between-group effects but were sufficiently powered to detect within-person effects and within- and between-person interactions. Of note, the main effect of interest in these analyses was the interaction between EDA over time (a within-person effect) and ASSC (a between-person effect), an effect that was sufficiently powered to detect a large effect.

## 3. Results

*Descriptive Statistics and Correlations*

The reporting of these results complies with STROBE guidelines for observational research. A total of 36 participants completed the study, with 17 participants (47.2%) being assigned to the niacin condition. Participants were 69.4% female. Age data were missing for 28 participants due to an administrative error. The mean age of the remaining participants was consistent with the age of undergraduate students at Ohio University ($M = 18.9$, $SD = 0.84$). Of the 36 participants, two were missing ASSC data (one in the niacin condition) and three were missing subjective distress ratings (two in the niacin condition), resulting in a sample of 31 for analyses of self-report data. The distribution of questionnaire responses can be found in Figure 1. ASI-3 social concerns and API scores were within acceptable ranges on metrics of normality based on simulation studies (i.e., skew $\leq$ |2|, kurtosis $\leq$ |7|; [33]). Sensitivity analyses were conducted to determine if the outlying scores on the API influenced analyses including this measure by limiting analyses to individuals who reported scores <13 on the API. There were no substantive differences in analyses between this subset of the sample and the total sample; therefore, analyses including the entire sample are presented. Regarding EDA data, 11 had poor EDA data quality (six in the niacin condition) and nine were missing pre-speech EDA data (five in the niacin condition). A total of 16 participants were included for analyses of EDA. All participants responded correctly to two attention-check items embedded in the questionnaires. Descriptive statistics and correlations among all study variables can be found in Table 1.

A linear regression analysis was conducted to examine the effects of ASSC, drug condition, instruction condition, the ASSC by drug interaction, and the ASSC by instruction interaction on subjective distress reported during the speech. The ASSC by drug ($B = -0.06$, 95% CI [$-0.42$, 0.29], $p = 0.71$) and ASSC by instruction ($B = -0.00$, 95% CI [$-0.38$, 0.37], $p = 0.98$) interactions were nonsignificant, so a final model was estimated without the interaction terms (see the left panel of Table 2). Overall, the predictor variables explained a significant amount of variance in subjective distress ($F$ [3, 27] = 2.93, $p = 0.05$; $\Delta R^2 = 0.25$). ASSC was related to subjective distress after controlling for drug and instruction conditions ($B = 0.15$, 95% CI [0.01, 0.29], $p = 0.04$). Drug condition ($B = 0.39$, 95% CI [$-1.18$, 1.95], $p = 0.62$) and instruction condition ($B = -0.94$, 95% CI [$-2.51$, 0.64], $p = 0.23$) were not significantly related to subjective distress after controlling for ASSC.

**Table 1.** Descriptive Statistics and Correlations.

| Variables | ASSC | Distress | Post Panic | Pre EDA | Early EDA | Pause EDA | Late EDA |
|---|---|---|---|---|---|---|---|
| ASSC | – | | | | | | |
| Distress | 0.43 * | – | | | | | |
| Post Panic | 0.35 * | 0.50 ** | – | | | | |
| Pre EDA | 0.22 | 0.24 | 0.31 | – | | | |
| Early EDA | 0.06 | 0.27 | 0.09 | 0.96 *** | – | | |
| Pause EDA | 0.04 | 0.27 | 0.10 | 0.94 *** | 0.99 *** | – | |
| Late EDA | 0.02 | 0.31 | 0.15 | 0.93 *** | 0.98 *** | 0.99 *** | – |
| Mean (%) | 11.06 | 5.08 | 6.94 | 8.72 | 9.46 | 9.41 | 9.19 |
| *SD* | 5.40 | 2.19 | 8.69 | 4.87 | 5.42 | 5.37 | 5.43 |

Note. *n*'s = 15–31. ASSC = Anxiety sensitivity social concerns. EDA = Electrodermal activity. *** $p < 0.001$, ** $p < 0.01$, * $p < 0.05$.

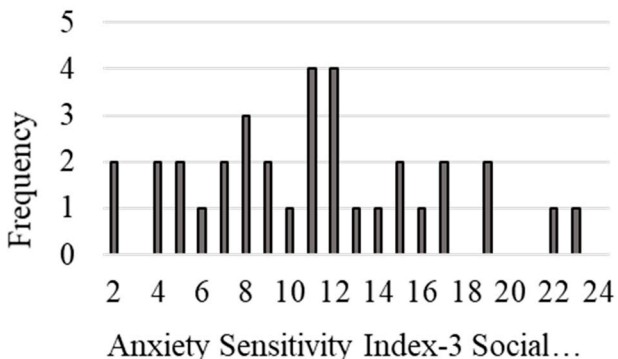

**Figure 1.** Questionnaire response distributions.

**Table 2.** Linear Multiple Regression Models.

| Effect | Subjective Distress | | | | | Panic Symptoms | | | | |
|---|---|---|---|---|---|---|---|---|---|---|
| | *B* | *SE* | 95% CI | | *p* | *B* | *SE* | 95% CI | | *p* |
| | | | LL | UL | | | | LL | UL | |
| Intercept | 5.51 | 0.72 | 4.02 | 6.99 | <0.001 | 6.14 | 3.05 | −0.08 | 12.36 | 0.05 |
| ASSC | 0.15 | 0.07 | 0.01 | 0.29 | 0.04 | 0.52 | 0.29 | −0.06 | 1.11 | 0.08 |
| Instructional Set | −0.94 | 0.77 | −2.51 | 0.64 | 0.23 | −0.05 | 3.19 | −6.56 | 6.46 | 0.99 |
| Drug Condition | 0.39 | 0.76 | −1.18 | 1.95 | 0.62 | 2.39 | 3.12 | −3.99 | 8.76 | 0.45 |
| ASSC × Instructional Set | – | – | – | – | – | – | – | – | – | – |
| ASSC × Drug Condition | – | – | – | – | – | – | – | – | – | – |

Note. Drug Condition dummy coded such that 1 = niacin condition. Instructional Set condition dummy coded such that 1 = increased salience condition.

A linear regression analysis was conducted to examine the effects of ASSC, drug condition, instruction condition, the ASSC by drug interaction, and the ASSC by instruction interaction on panic symptoms reported following the speech. The ASSC by drug (*B* = −0.42, 95% CI [−1.78, 0.94], *p* = 0.53) and ASSC by instruction (*B* = 0.39, 95% CI [−1.10, 1.89], *p* = 0.59) interactions were nonsignificant, so a final model was estimated without the interaction terms (see the right panel of Table 2). Overall, the predictor variables did not explain a significant amount of variance in panic symptoms (*F* [3, 30] = 1.60, *p* = 0.21; $\Delta R^2$ = 0.14). ASSC was marginally related to panic symptoms after controlling for drug and instruction conditions (*B* = 0.52, 95% CI [−0.06, 1.11], *p* = 0.08). Drug condition (*B* = 2.39, 95% CI [−3.99, 8.76], *p* = 0.45) and instruction condition (*B* = −0.05, 95% CI [−6.56, 6.46], *p* = 0.99) were not significantly related to subjective distress after controlling for ASSC.

A repeated-measures ANOVA was then conducted to examine the effects of ASSC, drug condition, instruction condition, the ASSC by drug interaction, and the ASSC by instruction interaction on EDA across pre-speech, early-speech, pause, and late-speech time windows. The three-way interaction between ASSC, drug, and time and the three-way interaction between ASSC, instructions, and time were nonsignificant, so they were not included in the final model. EDA signals did not differ over time (*F* [3, 33] = 0.12, *p* = 0.95, partial $\eta^2$ = 0.01). There was a significant within-person ASSC by time interaction (*F* [3, 33] = 5.40, *p* = 0.004, partial $\eta^2$ = 0.33) such that ASSC was more positively related to EDA during the pre-speech period relative to during the speech. The effect of ASSC on EDA was also qualified by a significant between-person ASSC by condition interaction (*F* [1, 10] = 5.11, *p* = 0.05, partial $\eta^2$ = 0.34) such that ASSC was positively related to EDA in the niacin condition but unrelated to EDA in the placebo condition.

### 4. Discussion

As hypothesized, ASSC predicted subjective distress during the niacin biological challenge; however, the influence of ASSC on subjective distress was not dependent on

condition. In the Allan et al. [12] study, ASSC was unrelated to subjective distress captured following the speech task. These discrepant results could be explained by when subjective distress was assessed in the current study. Unlike the previous niacin biological challenge study, subjective distress was assessed *during* the speech as opposed to *following* the speech. Thus, ASSC may amplify momentary distress during social stressors, but have an attenuated effect on distress after the stressor has passed. However, this may be partially due to the inclusion of participants with normative levels of ASSC. Additionally, ASSC predicted panic symptoms following the speech task. Contrary to Allan et al. [12], this effect was not stronger in the niacin condition. Together, results indicate that the niacin biological challenge task triggers distress across multiple modalities, particularly in individuals with elevated ASSC. The current study also was designed to assess the impact of ASSC on EDA throughout the speech task in an extension of the solely self-report methods of the Allan et al. [12] study. As hypothesized, ASSC positively predicted EDA signals throughout the speech task in the niacin condition but not the placebo condition. This finding is the first to indicate that ASSC impacts physiological arousal during social stressors in addition to subjective distress during these tasks, and is consistent with anxiety sensitivity generally predicting physiological arousal during other biological challenge paradigms (e.g., $CO_2$ inhalation; [34]). Moreover, the finding that ASSC only predicted EDA in the niacin condition is an indication that ASSC amplifies physiological markers of anxiety following the manipulation of facial flushing via niacin consumption.

Results from the current study also provide preliminary evidence for how ASSC impacts the time course of physiological arousal upon initiating laboratory-based social stress tasks. Indeed, ASSC was more strongly related to EDA signals during the pre-speech period. This finding was contrary to expectations, as it was hypothesized that ASSC would compound physiological arousal throughout speech task, leading to a positive feedback loop that resulted in higher arousal at the end of the speech task. However, the opposite effect was observed in the current study. Thus, ASSC may have a particularly strong impact on psychological distress anticipating entry into social situations. However, it should be noted that undergraduate students were recruited based on overall levels of anxiety sensitivity, rather than levels of ASSC. In general, undergraduate students have lower severity of anxiety-related symptoms and impairment than treatment-seeking community members. It may be that ASSC differentially impacts the time course of physiological arousal in social situations among participants with pathological levels of social anxiety or high levels of ASSC. In particular, it may be that individuals with normative levels of social anxiety would habituate during the niacin biological challenge; whereas participants with pathological levels of social anxiety would fail to habituate during this task.

The current study provides a template for investigations using the niacin biological challenge paradigm and elucidates avenues for future research. This study expanded upon the Allan et al. [12] study by integrating EDA as a physiological measure of fear reactivity. Future research may benefit from including additional physiological metrics. In particular, using photoplesmyography to measure facial blood flow in response to the niacin challenge would increase understanding of the impact of ASSC on physiological signals during social evaluative tasks [35,36]. The inclusion of photoplesmyography would also provide a manipulation check, allowing researchers to test whether facial blood volumes differ between the niacin and placebo conditions.

The results of the current study have clinical implications. In particular, the finding that ASSC most strongly impacts EDA during the anticipatory period leading to a social stressor may elucidate a target for intervention for patients with elevated ASSC. Clinicians may benefit from employing strategies to combat anticipatory anxiety related to observable symptoms to ultimately reduce the impact of ASSC and related sequelae. Finally, ASSC having a reduced impact on EDA over the course of the speech provides support for the central tenet of exposure therapy, namely that by facing feared situations, anxiety will decline over time [37]. Indeed, in the current study, physiological arousal (EDA signals)

declined throughout the speech task, showing habituation to this stress test, albeit in a university sample.

There are several limitations of the current study. First, the present study used a small, undergraduate sample, which may limit the generalizability of the current findings and highlight the need for replication in larger, more diverse samples. Null results involving EDA should be interpreted with caution, given the small number of participants included in these analyses. Despite the small sample size, several significant effects were observed in the expected direction, indicating that the sample was sufficiently powered to detect some of the central effects. Further, although all participants had at least average overall AS scores on the ASI-3 ($M$ = 11.06 in the current study versus $M$ = 17.3 in clinical samples with SAD) [3] participants had relatively low ASSC scores relative to clinical samples. The small sample, in combination with the relatively low ASSC scores relative to clinical samples likely contributed to some of the nonsignificant effects that were significant in the Allan et al. [12] study. To increase confidence in the results, replication is needed in larger samples, preferably of participants with elevated social anxiety. Additionally, several unmeasured third variables may impact the internal validity of this study. For example, prior research has suggested that culture and various demographic characteristics influence the presentation, prevalence, and treatment of social anxiety [38]. However, limited information regarding demographic and cultural background was assessed in this study. Thus, essential to replication efforts is the need to assess the influence of demographic variables on the relations between ASSC and responding to the niacin biological challenge to determine if these findings can generalize across demographic and cultural groups. Finally, participants were told to give a speech on their greatest weakness in the current study. It may be that this speech prompt was not sufficiently stressful to activate an anxiety response in all participants, particularly given the current sample consisted of undergraduate students (resulting in an average subjective distress rating of 5.08 on a 1–10 scale). Future studies may benefit from including a more emotionally salient speech topic.

## 5. Conclusions

The current study, in combination with the Allan et al. [12] study, provides support for the niacin biological challenge as a laboratory-based social stress task to investigate processes contributing to anxiety following the manipulation of facial flushing. Across studies, ASSC has emerged as a predictor of subjective and objective markers of distress during and following this task, suggesting ASSC consistently amplifies anxious responding following the onset of publicly observable anxiety symptoms. Indeed, this study is the first to show that ASSC impacts physiological signs of arousal during the niacin biological challenge. By measuring additional physiological markers of anxiety during the niacin biological challenge paradigm, researchers will have a more complete understanding of the processes through which ASSC contributes to acute anxiety in social situations. Together, these results suggest the niacin biological challenge is a valid paradigm to assess social anxiety following the onset of facial flushing and implicate ASSC as an amplifier of distress during this task.

**Author Contributions:** Conceptualization, K.G.S. and N.P.A.; data curation, K.G.S.; formal analysis, K.G.S.; investigation, K.G.S. and N.P.A.; methodology, K.G.S. and N.P.A.; project administration, K.G.S. and N.P.A.; resources, N.P.A.; supervision, N.P.A.; writing—original draft, K.G.S. and M.V.; writing—review and editing, K.G.S., M.V. and N.P.A. All authors have read and agreed to the published version of the manuscript.

**Funding:** This study was conducted using startup funds available to Dr. Allan from Ohio University.

**Institutional Review Board Statement:** All subjects gave their informed consent for inclusion before they participated in the study. The study was conducted in accordance with the Declaration of Helsinki, and the protocol was approved by the Institutional Review Board at Ohio University (protocol number: 16-F-40).

**Informed Consent Statement:** Informed consent was obtained from all subjects involved in the study.

**Data Availability Statement:** Not applicable.

**Conflicts of Interest:** The authors declare no conflict of interest.

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
