# Peer review of "Anxiety Sensitivity Social Concerns Predicts Electrodermal Activity during the Niacin Biological Challenge Paradigm"

_2673-5318, doi:10.3390/psychiatryint3040028_

Round 1
Reviewer 1 Report (Previous Reviewer 3)
Thank you very much for returning this manuscript. Although the authors have tried to explain their choices, the manuscript still lacks evident validity.
For example, the sample size without any population parameter still does not allow generalizing the findings.
The lack of validity of the instruments also affects the internal validity of the manuscript. Saying that it was applied in English-speaking countries does not diminish this bias.
Despite the authors' considerations about the validation (CONTENT FOR THE ENGLISH LANGUAGE) of the instrument, with citation, the arguments are still fragile. The validity and/or calibration of an instrument is conditioned not only to the content validation (as the authors did), but also to the number of times it is tried and the analytical procedures, including multivariate statistics. There are several studies that reinforce the need for combination with more consistent and robust analyses, such as criterion and construct, in order to gather more evidence on validity. Reliability is also something that should be added to the results: See: Souza AC, Alexandre NMC, Guirardello EB. Psychometric properties in instruments evaluation of reliability and validity. Epidemiol Serv Saude. 2017 Jul-Sep;26(3):649-659. English, Portuguese. doi: 10.5123/S1679-49742017000300022. PMID: 28977189; https://doi.org/10.1590/0102-311X00143613; Reichenheim ME, Hökerberg YH, Moraes CL. Assessing construct structural validity of epidemiological measurement tools: a seven-step roadmap. Cad Health Publica. 2014 May;30(5):927-39. doi: 10.1590/0102-311x00143613. PMID: 24936810. Content validation is an important step, but it must be added to other procedures and even to a more representative sample for the convergence of objectives, that is, that the instrument measures what it really intends and or was intended to measure.
Author Response
Please see the attachment.

Reviewer 2 Report (Previous Reviewer 2)
no comment
Author Response
No comments.
Reviewer 3 Report (Previous Reviewer 1)
I thank the authors for the revision of their manuscript. I acknowledge that they have improved the theoretical background of their work and they have also added important limitations. However, the study has serious flaws that challenge its validity and merit. The study is severely underpowered.
Additionally, I have some queries:
Methods:
The authors should explicitly mention what the study design is.
In the exclusion criteria, the authors should mention how many participants were excluded. Additionally, they should specify how these criteria were measured, if it is by self-report of the participant, they should mention it.
The authors should describe in detail the post-hoc power analysis used to estimate the sample size.
In instruments, the authors should explicitly show the most relevant validation values.
In the analysis plan section, authors should sort the paragraph by: descriptive (missing), bivariate (hypothesis testing), and multivariate (regression analysis). Additionally, more detail on missing data handling is required.
Results:
The following sentence should be rewritten "The mean age of the remaining participants was consistent with other studies including undergraduate samples (M = 18.9, SD =.84; e.g., Geyer et al., 2018; Kawamura et al., 2001)." In the results section, comparisons with results from other articles cannot be made.
Of concern is the missing data on the age variable, coupled with the very small sample size of the research.
The authors should add the confidence intervals of the regression analysis.
I suggest adding a figure of the distribution of responses to the two questionnaires and a regression analysis table.
Round 2
Reviewer 1 Report (Previous Reviewer 3)
The validity problems still remain. The authors seem not to understand validation well, and confuse a series of parameters placed in the opinion. I suggest that the study be redone or at least present a test of sample power and studies and evidence that attest to the real validity of the instruments
Author Response
Please see attached response.

Reviewer 3 Report (Previous Reviewer 1)
I congratulate the authors for improving the manuscript with the suggestions/comments submitted.
Author Response
We appreciate the reviewer's time spent reviewing this manuscript and their thoughtful critiques.
This manuscript is a resubmission of an earlier submission. The following is a list of the peer review reports and author responses from that submission.
Round 1
Reviewer 1 Report
Saulnier et al. aimed to examine the effects of ASSC on self-reported distress and electrodermal activity (EDA) during the niacin biological challenge in a sample of undergraduates.
It is an interesting article but there are some shortcomings that need attention. I would like to make some suggestions and contributions to the present manuscript. The authors need to check that their manuscript complies with STROBE guidelines for observational research: https://www.equator-network.org/reporting-guidelines/strobe/ A statement needs to be added to the manuscript confirming the same.
Introduction:
The purpose of the article is clearly presented. Overall this is well-written and puts the study in context with reference to up-to-date and relevant literature, however, strengthening the rationale is likely to improve the manuscript. Authors should describe gaps or limitations in the literature: lack of studies, studies in non-generalizable contexts, and methodological deficiencies. In addition, indicate how your analysis will avoid these limitations. In your introduction, you should discuss the potential usefulness of answering the research question.
Methods:
The information shown in lines 103-107 should be included in the "Results" section. The information shown in lines 107-110 corresponds to section 2.4. Ethical aspects in "Methods".
The total study population should be mentioned. Specify the calculation used for the sample size and sampling techniques used. In addition, the selection criteria (inclusion and exclusion) are not mentioned. The authors should add a participant selection flowchart.
In Instruments, although reliability information is included, no data on the validity of the questionnaires are shown.
Describe in more detail the data collection process.
The statistical analysis plan and ethical aspects for this research are not described.
Results:
It is not possible to evaluate exactly, since I do not know the statistical tests used for their analysis. It is important that you describe your analysis plan in detail in the Methods section.
Discussion:
The authors need to improve their discussion.
In limitations, authors should add information bias, mentioning if there are any confounding variables that have not been measured in their research. Additionally, discuss the generalisability (external validity) of the study results. In addition, in the limitations described, it is not enough to mention each one of them, but they should mention what was the potential solution to deal with these limitations and explain why their presence does not invalidate their study results.
Additionally, I suggest strengthening your discussion with the redaction of an "Implications of findings in mental health policy" paragraph.
Conclusions:
The authors should explicitly add a paragraph on the conclusions of their study, which should be logically validated and justified by the evidence of their findings.
References:
There were 25 and all are relevant and appropriate.
Reviewer 2 Report
1-why is there no equality between the numbers of males and females in the experiments?
2- why age is not uniform?
3- material and method, results are not enough
Reviewer 3 Report
I reviewed the manuscript entitled "Anxiety Sensitivity Social Concerns Predicts Electrodermal Activity During The Niacin Biological Challenge Paradigm".
The manuscript has a good degree of originality, interesting methods, clear results and high-level scientific writing.
But some details need to be clarified:
1. What are the population parameters for defining the sample? Where is the sample calculation? what kind was it? Without this information the results are unfeasible.
2. Have Anxiety Sensitivity Index-3 (ASI-3) and Acute Panic Inventory (API) been validated for your country? Where is the data?
3. The details of blinding and randomization are unclear.
4. The authors do not even include a data analysis section, and this is inappropriate. How was the data analyzed? for which tests? Why reasons?